# An electron counting algorithm improves imaging of proteins with low-acceleration-voltage cryo-electron microscope

Dongjie Zhu[1,2], Huigang Shi [2,3], Chunling Wu[2,3] & Xinzheng Zhang [2,3 ✉]

Relative to the 300-kV accelerating field, electrons accelerated under lower voltages are potentially scattered more strongly. Lowering the accelerate voltage has been suggested to enhance the signal-to-noise ratio (SNR) of cryo-electron microscopy (cryo-EM) images of small-molecular-weight proteins (<100 kD). However, the detection efficient of current Direct Detection Devices (DDDs) and temporal coherence of cryo-EM decrease at lower voltage, leading to loss of SNR. Here, we present an electron counting algorithm to improve the detection of low-energy electrons. The counting algorithm increased the SNR of 120-kV and 200-kV cryo-EM image from a Falcon III camera by 8%, 20% at half the Nyquist frequency and 21%, 80% at Nyquist frequency, respectively, resulting in a considerable improvement in resolution of 3D reconstructions. Our results indicate that with further improved temporal coherence and a dedicated designed camera, a 120-kV cryo-electron microscope has potential to match the 300-kV microscope at imaging small proteins.

[1] School of Life Sciences, Division of Life Sciences and Medicine, University of Science and Technology of China, 230026 Hefei, China. [2] National Laboratory of Biomacromolecules, CAS Center for Excellence in Biomacromolecules, Institute of Biophysics, Chinese Academy of Sciences, 100101 Beijing, China. [3] University of Chinese Academy of Sciences, 100049 Beijing, China. ✉email: xzzhang@ibp.ac.cn

During imaging, the exposure of proteins to accelerating electrons leads to irreversible and accumulative radiation damage[1–4]. Consequently, the total electron dose used in imaging proteins is limited, resulting in elevated shot noise, which yields a low signal-to-noise ratio (SNR) in cryo-electron microscopy (cryo-EM) images. A reconstruction of the 3D structure of a protein employs two steps: alignment and averaging[5]. A high SNR of the cryo-EM image is crucial for alignment accuracy, which then permits an effective averaging that increases the SNR in the 3D reconstruction, equivalent to improving the resolution of the reconstruction in cryo-EM. The signal from proteins in cryo-EM images is correlated with its size. Small proteins scatter less electrons, thereby producing weak signals, leading to low SNR images and therefore preventing accurate alignments[6,7]. From the perspective of SNR, the difficulty in reconstructions increases with decreasing protein size. Therefore, improving the SNR of cryo-EM images is extremely helpful in the reconstruction of small proteins.

The power of cryo-EM in resolving the 3D structure of proteins increased rapidly since 2013 through the application of direct detection devices[8,9] (DDDs). The DDD has high frame rate and improved detective quantum efficiency (DQE) compared with those of charge-coupled devices (CCDs) and films, allowing the recording of cryo-EM images with better SNR[10], helping to set off the 'resolution revolution'[11]. Currently, the state-of-the-art cryo-EM technique enables the reconstruction of proteins with sizes as low as 40 kD[12] but requires an ideal buffer condition and a thin layer of ice to minimise background noise apart from shot noise. However, in reality, an ideal cryo-EM specimen is difficult for many proteins. Consequently, any method that further improves the SNR of cryo-EM images benefits the structural determination of small proteins.

The power of scattered electrons by an atom is negatively correlated with the energy of the incident electron[13]. Therefore, low-acceleration-voltage electrons produce more signal upon their interaction with the cryo-EM specimen. However, low-energy electrons create severer radiation damage[14], allowing fewer electrons for imaging, which may compromise the overall SNR of a cryo-EM image. To measure the dependence of the radiation damage rate on the electron energy, Peet et al.[15] measured the inelastic scattering cross-section of electrons from graphene and the radiation damage ratio of electrons with energies between 100 kV and 300 kV from 2D-crystals of bacteriorhodopsin and paraffin. Instead of using the measured inelastic scattering section, they used Estar database[16] to calculate the information coefficient (IFC). Their results indicate an optimal voltage of 100 kV for 30-nm-thick samples, with the signal improving by 5% compared with that at 300 kV. For 10 nm-thick samples, their result indicates a 25% improvement in signal by lowering the acceleration voltage to 30 kV. However, the radiation damage estimated from the critical dose was notably different between the crystal sample and the vitrified non-crystal sample[17]. As a result, the radiation damage measured from the 2D-crystals may be inaccurate and leads to a different optimal voltage. Therefore, an accurate electron energy-dependent radiation damage rate for cryo-EM specimens remains undetermined.

The currently available DDD cameras for cryo-EM images were designed for detecting 300-kV electrons[18]. Their DQE drops with decreasing acceleration voltage[19]. Counting algorithms[20,21] have been developed to determine the position of the electron entering the camera and to normalise its intensity. Some DDDs implement a counting algorithm that calculates the position of the centre of mass/gravity for each signal as the position of the incident electron[19]. The accuracy of this position has been shown sometimes at a sub-pixel level[22]. If the column charge from the incident electron is isotropic, the centre-of-mass position is an effective approach in determining the incident position. However, if the incident electron is back-scattered or stimulates numerous of pixels, the centre-of-mass position may diverge from the true incident position.

In contrast, a hybrid pixel camera[23–26] (HPC) has a higher DQE for lower energy electrons and has the potential to replace the DDD in low-acceleration-voltage cryo-EM. However, current HPCs have significantly larger physical pixel size than DDD cameras, meaning fewer pixels on a single chip. Because of the large physical pixel size of the HPC and low DQE at 300 kV[27], the low data throughput limits the usage of the HPC in cryo-EM. For instance, one HPC[26] that was used for recording 100-kV cryo-EM images had only approximately 3% of the number of pixels in a current 4k-DDD because its physical pixel size is 75 μm, producing only 3% of the throughput during data collection with the same magnification. The advantage of throughput in DDDs suggests that improvements in the detection efficiency of low-energy electrons for existing DDD cameras is valuable and remains to be investigated.

Apart from that, the low-acceleration voltage intensifies the Ewald sphere effect and chromatic aberration[28]. Different reconstruction algorithms have been successful in correcting the Ewald sphere effect[29–31]. However, whether signal depression by chromatic aberration is a limiting factor in preventing the low-acceleration-voltage cryo-electron microscope from achieving pseudo-atomic resolution remains to be investigated.

Here, using the single particle method, we calculate the critical doses from different datasets recorded from the same vitrified protein sample with different acceleration voltages. Our results show that the radiation damage rate in the vitrified protein samples is faster than that in 2D-crystals and fits the inelastic scattering section described by an empirical equation or Estar database. We develop an electron counting algorithm for DDDs. When implemented in a Falcon III camera, we are able to gain 8% and 20% SNR improvement in counting 200-keV and 120-keV electrons at the 0.5-time Nyquist frequency and approximately 21% and 80% SNR improvement at the Nyquist frequency, respectively. The improvement in SNR for the DDD in detecting 120-keV electrons enhances the accuracy of alignment and significantly improves the structure resolution of proteins at 120 kV. By estimating signal suppression being introduced from chromatic aberration, our results indicate that chromatic aberration is a major factor that limits the resolution of the 3D reconstruction from 120-kV cryo-EM.

## Results

**Acceleration voltage-dependent critical dose and optimal voltage of vitrified protein samples of various sizes.** To test whether low-accelerate voltage cryo-EM produces more signal than widely-used 300-kV cryo-EM as suggested by Peet et al.[15], we measured critical doses[17] on a 30-nm-diameter Coxsackievirus A10 sample at three voltages: 300, 160 and 120 kV. Because the TEM we used lacked of Faraday cup on the sample side, several calibrations had to be performed (Fig. 1b, Method, Supplementary Figs. 1–2, Supplementary Note 1–4) before we measured the critical doses relative to that of 300 kV (Fig. 1c). The relative radiation damage per-electron at 300 kV, 160 kV and 120 kV are 1.00, 1.38 and 1.57 (Fig. 1d). Comparing our result with that of Peet et al.[15], we show that with decreasing acceleration voltage, the radiation damage per-electron is severer in the vitrified protein samples than in crystalized bacteriorhodopsin and paraffin. Our data falls in between the lines plotted by data from Estar database and an empirical function by Wall's et al.[32] (Fig. 1d).

Our results confirmed that putting the effects of elastic scattering, inelastic scattering and radiation damage together,

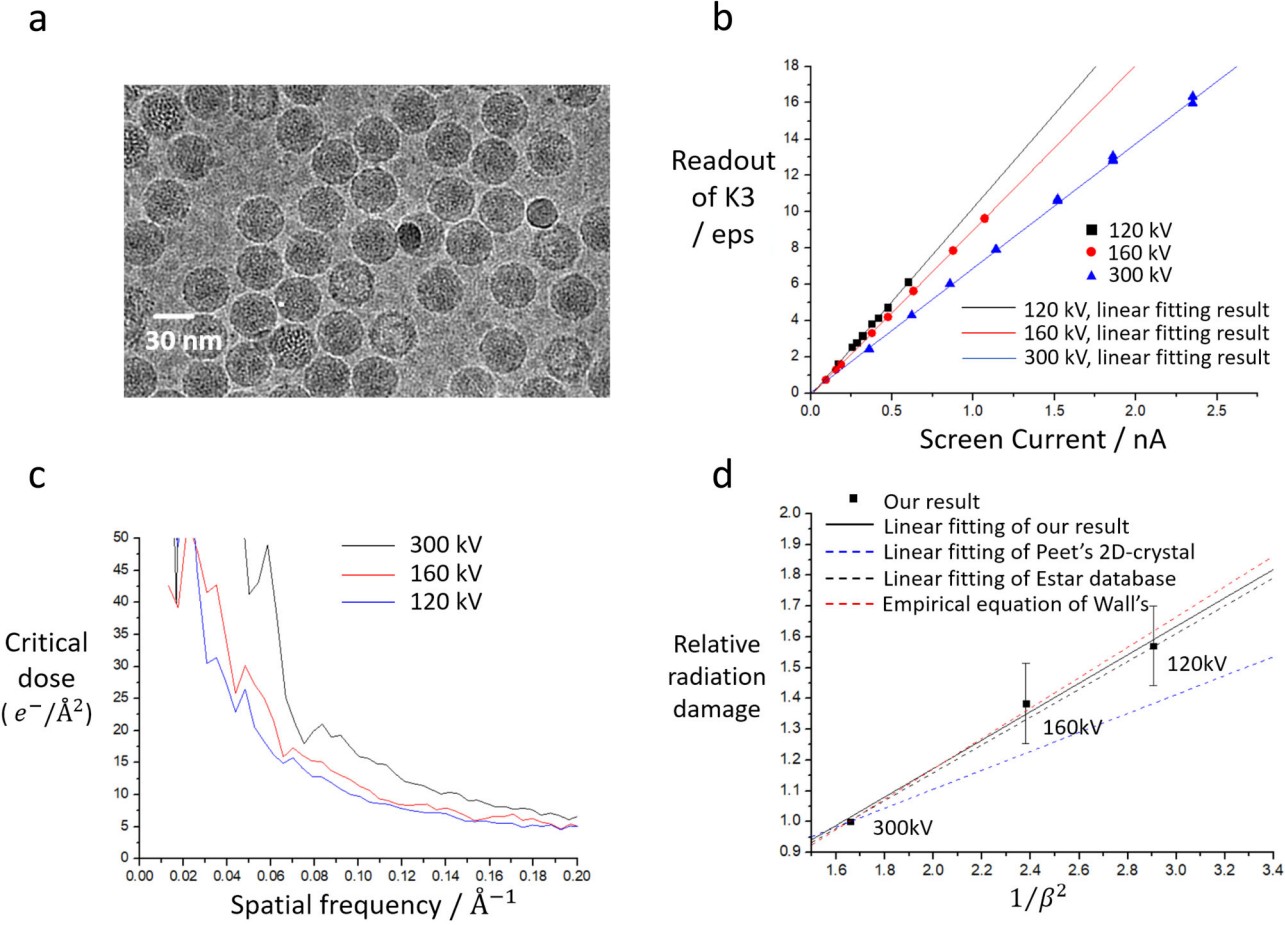

**Fig. 1 Measuring radiation damage in vitrified protein. a** Sample image of Coxsackievirus A10. Image was taken at 120 kV with defocus of −1.0 μm. Sigma-contrast of image was set to three and low-passed to 20 Å. **b** Comparison of raw-readouts of K3 camera to screen current at 300/160/120 kV. The K3 operated on non-CDS mode and magnification of 22,500x. **c** The critical doses of Coxsackievirus A10 at 300/160/120 kV. The critical doses were calibrated by 'counting scaling factor' and rescaled to fit Grant's results at 300 kV. **d** Comparison of critical doses of vitrified protein samples to 2D-crystal samples. The relative radiation damage uses critical dose of 300 kV as a reference and obtained by averaging the results in frequency between 10 Å to 5 Å of Fig. 1c. Error bars represent 1 standard deviation. $\beta$ represents the ratio between electron velocity and speed of light. The linear fitting result is $y = 0.24832 + 0.46211/\beta^2$., with R-square = 0.97823. The 'Empirical equation' line was a linear fitting result from function of Wall's[32]. The 'Estar database' line was the linear fitting result of stopping-power of carbon from Estar database[16].

the optimised energy of the incident electrons that produce the strongest signals is protein size dependent[15]. Since our results in radiation damage ratio being close but slightly higher to Estar database, our prediction to SNR improvement at 100-kV cryo-EM than 300-kV would be slightly weaker, ~3% to that ~5% from Peet et al. for 30-nm proteins.

**Size of electron signal on camera determines its SNR.** To understand the drop in DQE of a DDD detecting low-energy electrons, we examined the raw output frames from the Falcon III camera with SNR Tool software (Thermo Fisher Scientific). An incident electron usually stimulates a couple of neighbouring pixels. We defined these pixels as a 'cluster'. A typical raw image at 300 kV and 120 kV was acquired by the Falcon III camera before counting and the corresponding clusters were calculated (Method, Supplementary Note 5 and Supplementary Fig. 3a).

After detecting each cluster that represented a single-electron event, we calculated the percentage of clusters stimulated by the 120-keV, 200-keV and 300-keV electrons according to number of pixels in a cluster (Fig. 2a). The average sizes of cluster were 3.7, 2.7 and 2.5 pixels for the 120-keV, 200-keV and 300-keV electrons, respectively, similar to the results in previous study[19]. We divided all clusters into three groups (Fig. 2a) according to

size: 'small' for 1 and 2 pixels, 'medium' for 3 and 4 pixels, and 'large' for 5 or more pixels (Examples shown in Supplementary Fig. 3b). The reason for this division is to be given later. At 120-kV acceleration voltage, the small/medium/large-sized clusters contain 33.8%/38.6%/27.6% of the total, 59.0%/30.2%/10.8% at 200 kV, and 60.2%/31.9%/7.9% at 300 kV. On each image, the small/medium/large-sized clusters were selected to form three images, image$_{small}$, image$_{medium}$, and image$_{large}$, respectively.

We tried to measure the accuracy of the electron position given by a home-made counting algorithm Mass-Centre filter (MCF) that calculate mass-centre of a cluster in super-solution mode on image$_{small}$, image$_{medium}$, and image$_{large}$ (Method and Supplementary Note 6). The accuracy of the electron position can be measured from the DQE (Method). However, the result produced by FINDDQE[33] sometimes varied considerably and were unusable for a stable determination of the SNR[34].

Instead of measuring the DQE, we collected four cryo-EM datasets: two on Coxsackievirus A10 at 120 kV and 200 kV, and two on apo-ferritin at 120 kV and 300 kV and we examined the quality of counting images based on the gold-standard Fourier shell coefficient (FSC) curves of the reconstructions. For each image in the Coxsackievirus A10 dataset at 120 kV, 7 images with cluster sizes varying from 1 to 7 pixels that containing roughly the

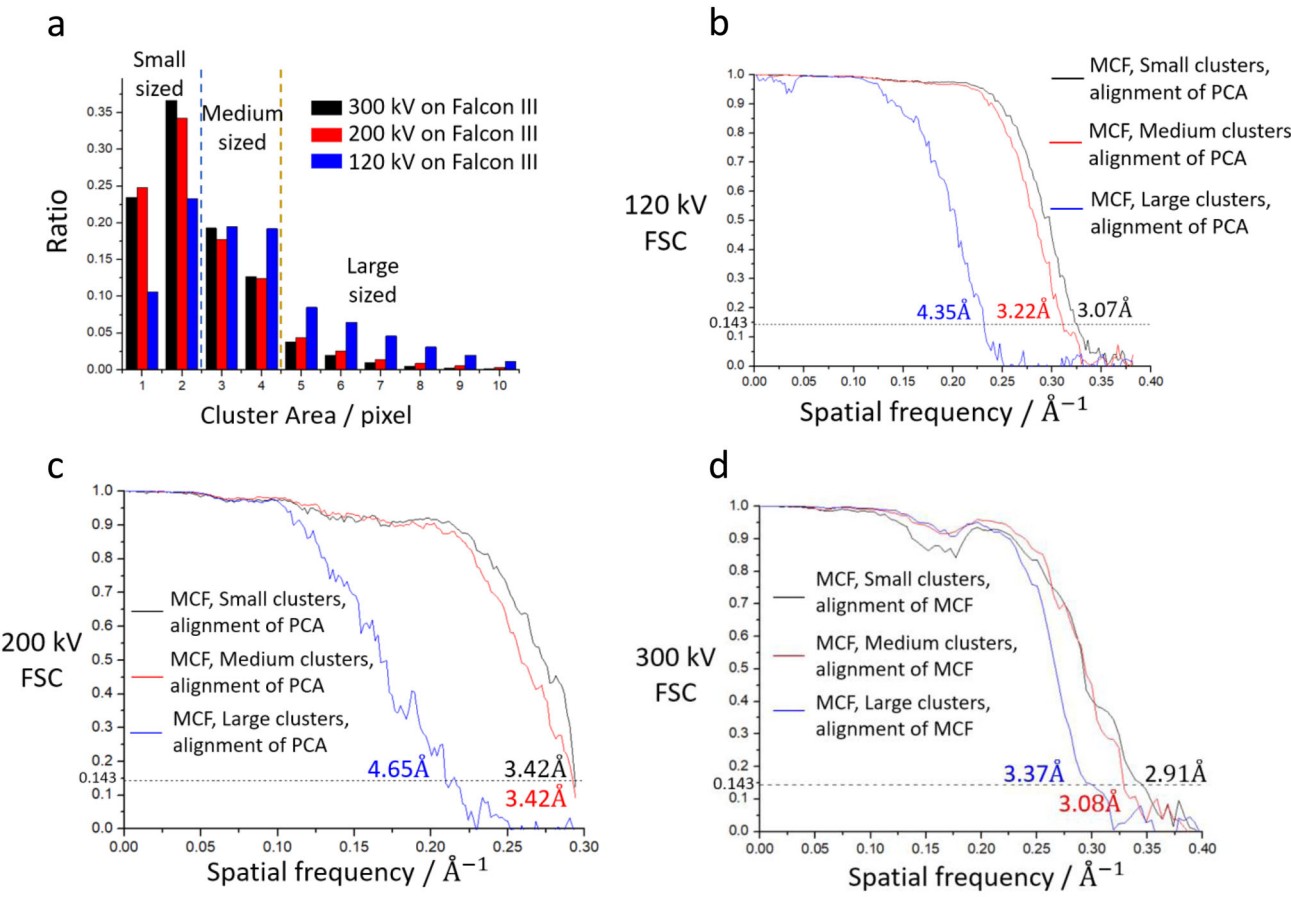

**Fig. 2 Sizes of clusters and corresponding FSCs of Mass-Centre filter (MCF) algorithm. a** The relative ratio with the same pixel of cluster area. 'Small-sized', 'medium-sized' and 'large-sized' clusters are distinguished by blue and brown dash lines. **b** FSC curves of Coxsackievirus A10 at 120 kV with MCF counting algorithm. Each pair of reconstructions holds roughly the same number of clusters (~27% of total). PCA is short for Pattern Counting Algorithm we proposed. **c** FSC curves of Coxsackievirus A10 at 200 kV with MCF counting algorithm. Each pair of reconstructions holds roughly the same number of clusters (~10.8% of total). **d** FSC curves of apo-ferritin at 300 kV with MCF counting algorithm. Each pair of reconstructions holds roughly the same number of clusters (~7.9% of total). All the resolution of reconstructions based on gold-standard FSC criterion marks on the graph.

same number of randomly selected electrons were produced using the MCF, which made 7 datasets to yield 7 reconstructions using the same alignment file generated by RELION, from the highest resolution reconstruction achieved latter in this work, with images produced by our improved counting algorithm. We calculated the gold-standard FSC curve for each reconstruction (Supplementary Fig. 4a). The FSC curves corresponding to cluster size of 1 and 2 pixels, and 3 and 4 are similar, whereas those of pixel areas of 5 pixels and more dropped rapidly at 7 Å and above. This is the main reason for dividing the clusters into three types.

Furthermore, we calculated $image_{small}$, $image_{medium}$, and $image_{large}$ from Coxsackievirus A10 dataset at 120/200 kV and apo-ferritin dataset at 300 kV with the same method. The FSCs of these three images were also calculated (Fig. 2b–d). As expected, the FSC curve of $image_{large}$ at 120 kV drops with increasing frequency much faster than $image_{medium}$ and $image_{small}$; $image_{small}$ was the best of the three. At 300 kV, the FSCs of $image_{medium}$ and $image_{small}$ were closer. The FSCs of $image_{large}$ was still worse than $image_{small}$ and $image_{medium}$, despite their gap narrowing.

**Comparing filters that transform cluster into a single point and applying SNR weighting.** The drop in the FSC for $image_{large}$ suggests that the location accuracy of mass centres from large-sized clusters is worse than that of small/medium clusters. Home-made

counting algorithms Peak-value filter (PVF) and Geometric-Centre filter (GCF) were used to determine peak-value position and geometric centre of a cluster (Supplementary Note 6), respectively. These clusters were used to re-produce counting images: $image_{small}$, $image_{medium}$ and $image_{large}$ (Method). FSC curves of the reconstructions in $image_{small}$, $image_{medium}$, and $image_{large}$ from PVF, GCF and MCF counting algorithm at 120 kV are shown separately in Supplementary Fig. 4b–d. At 120 kV, the GCF method is the worst of the three counting methods, and for the small and medium-sized clusters, the results obtained from the MCF and PVF methods are similar. Surprisingly, the PVF method produces markedly better results than the MCF method for the large-sized clusters, in that the resolution of reconstruction improves by ~0.7 Å.

Similarly, comparisons are presented about the PVF method and MCF at 200 kV (Supplementary Fig. 4e) and 300 kV (Supplementary Fig. 4f). GCF method was discarded because of its bad performance at 120 kV. The results of the MCF method are slightly better than those from the PVF method for small-sized clusters. Once again, the PVF method is better than the MCF method for large-sized clusters at 200 or 300 kV. At 200 kV, the gap between FSCs from small-sized clusters and large-sized clusters is still significant. In contrast, because of the lower ratio of large-sized clusters and a narrower gap between FSC of large-sized clusters and small-sized clusters for 300-keV electron, degradation of SNR due to the large-sized clusters in

the combined image is marginal compared with that for the 120 or 200-keV electrons.

From the results above, we found that the reconstructions from large-sized clusters we classified had low FSCs. As a fixed alignment file was used for all the reconstructions, the SNR of the three types of clusters could be directly calculated from the FSCs. When clusters were combined to form a cryo-EM image, weighting factors (Method and Supplementary Note 7–8) should be applied to clusters of different SNRs to increase the overall SNR of the full image.

Using the deduced weighting factor, a combination of the MCF method for small/medium-sized clusters and the PVF method for large-sized clusters with weighting (Weighted-point filter, or WPF) improved not only the SNR of images which was 2.5% at 0.5x Nyquist and 43.4% at Nyquist frequency by linear fitting (Supplementary Fig. 4h) but also the resolution of each reconstruction after refinement on the 120-kV dataset (Supplementary Fig. 4i).

**Counting algorithm with improved weighting based on the geometric shape of clusters**. Applying a weighting function to a counting image in reciprocal space is equivalent to convoluting the real-space image with the spreading function—the Fourier transform of the weighting function. After convolution, each electron event originally represented by a point extends to a homogeneous disk (Supplementary Fig. 5a). However, as shown in Supplementary Fig. 3b, each large-sized cluster has a unique shape. A spreading function that takes the shape of the cluster into account may be better.

To keep the shape of a cluster, all pixels within a cluster should be recorded as an image with the total sum of the pixel values within a cluster being normalised to one[35]. We propose a pattern counting algorithm (PCA, Supplementary Note 6) to do so. It contains an adjustable parameter *pow* to modify the pixel values by scaling each pixel value within a cluster before normalising (Supplementary Note 6). We tested the *pow* parameter for the small/medium/large-sized clusters from the Coxsackievirus A10 dataset at 120 kV (Method). Various of values of *pow* were tested (Method). To simplify the figures, we only present results from limited values of *pow* in the figures below.

For all clusters, the results (Supplementary Fig. 5b) show interestingly that *pow* = 1 is generally better, having a slight advantage at high frequency. For small or medium-sized clusters (Supplementary Fig. 5c, d), *pow* = 1 is similar to *pow* = 2 at high frequency and better than *pow* = 0. When compared with MCF, the *pow* = 1 performs equivalently or slightly better with small-sized clusters, and quite noticeably better with medium-sized clusters.

For large-sized clusters, every *pow* presents a much higher FSC than that with the PVF method (Supplementary Fig. 5e). *pow* = 0 is better than *pow* = 1 above 7 Å, corresponding to approximately 15% SNR boost at 0.5× the Nyquist frequency (Supplementary Fig. 5f). Combining all the test results of *pow*, we settled on *pow* = 1 to be used for small/medium-sized clusters, and *pow* = 0 for large-sized clusters at 120 kV. We also performed the same *pow* test for large-sized clusters in the Coxsackievirus A10 dataset at 200 kV and apo-ferritin dataset at 300 kV. The results (Supplementary Fig. 5g, h) show that the FSCs are almost identical for *pow* = 0, *pow* = 0.5, and *pow* = 1.0. We finally elected to apply *pow* = 1 for small/medium clusters and *pow* = 0 for large-sized clusters at 120/200/300 kV.

Because the PCA has a higher SNR than the MCF, we regenerated the PCA images in the 120 kV Coxsackievirus A10 dataset. In a comparison of the two, PCA images had more than 4 times SNR at 0.5× the Nyquist frequency for the large-sized clusters (Supplementary Fig. 6a), which helped increase the resolution from 4.35 Å to 3.27 Å (Fig. 3a) and the overall resolution from 3.05 Å to 2.95 Å (Fig. 3b), which is also being better than WPF using the same alignment file. For the combined images, the corresponding SNR improvement from the PCA method to the conventional MCF method (Fig.3c) is more than 80%, which was achieved at near Nyquist frequency, as established by the linear fitting result. At half Nyquist frequency, more than 20% SNR improvement was gain.

We also applied the PCA to the apo-ferritin dataset at 120 kV. The overall resolution was increased from MCF's 3.39 Å to PCA's 2.87 Å after refinement. It should be noted that motion correction, CTF measurement, classification and refinement were re-ran using the same script. As a result, their SNR cannot be compared directly. When we used the alignment file of the PCA images to reconstruct models from images obtained with MCF and WPF, the resolution was 2.90 Å and 2.87 Å, respectively (Fig. 3d). This result indicates that increasing the SNR in the PCA images led to an improved alignment. Because the final resolution of the PCA images reaches only 0.7x the Nyquist frequency, the resolutions of two reconstructions were close regardless of the counting algorithm we chose. For apo-ferritin at 300 kV, the PCA method fell behind the MCF method for combined images (Supplementary Fig. 6b), which indicated that MCF should be used at least for small or medium-sized clusters at 300 kV.

The PCA method can also be weighted using the SNR. In accordance with Eq. 3, the noise-power-spectrum (NPS) must be calculated separately for the three types of clusters (Supplementary Fig. 6c). For large-sized clusters, the PCA deploys a natural weight depressing the high frequency amplitude. Differently, the amplitude of small-sized clusters dropped much slower with the increasing of the frequency. The SNR weighting can still be applied to PCA despite the natural amplitude weight for large-sized clusters. After applying the SNR weighting on PCA data, the result shows only a marginal improvement after SNR weighting (Supplementary Fig. 6d, e), implying that SNR weighting of large-sized clusters in PCA is dispensable.

**Counting algorithms for the 120 and 200-kV datasets**. From the results above, deploying SNR weighting is necessary for the MCF method because it is able to provide an SNR increase of nearly 40% at the Nyquist frequency from WPF at 120 kV. Guo et al.[36] developed an event-based electron counting storage format—the Electron Event Representation (EER). This format only stores the position and time-stamp of each electron event recorded. For compatibility with the EER format, the WPF counting method we propose modifies the electron event record—a single bit is added to mark whether the electron to be SNR weighted.

A more complex but better counting method for 120-keV electron is the PCA method. To minimise the storage usage and maximise the SNR of small-sized clusters, we modified the PCA filter into a Hybrid method that combines MCF and PCA for clusters of different sizes. Hybrid method uses MCF for small and medium-sized clusters and a '*pow* = 0' PCA for large-sized clusters. No additional SNR weighting is used in Hybrid method. With Hybrid, we re-ran the SPA process using the two datasets at 120 kV. The final resolution for apo-ferritin was 2.87 Å; for Coxsackievirus A10, it was 2.88 Å (Fig. 4a).

For 200-kV dataset, we used $2 \times 2$ super-resolution mode of both Hybrid and MCF. After reconstruction, the final resolution of Hybrid was 2.66 Å, being 0.06 Å higher than MCF with the same alignment (Supplementary Fig. 6f). Corresponding SNR improvement is 8.0% at 0.5x Nyquist and 21.4% at 1.0x Nyquist frequency by linear fitting (Fig. 4b).

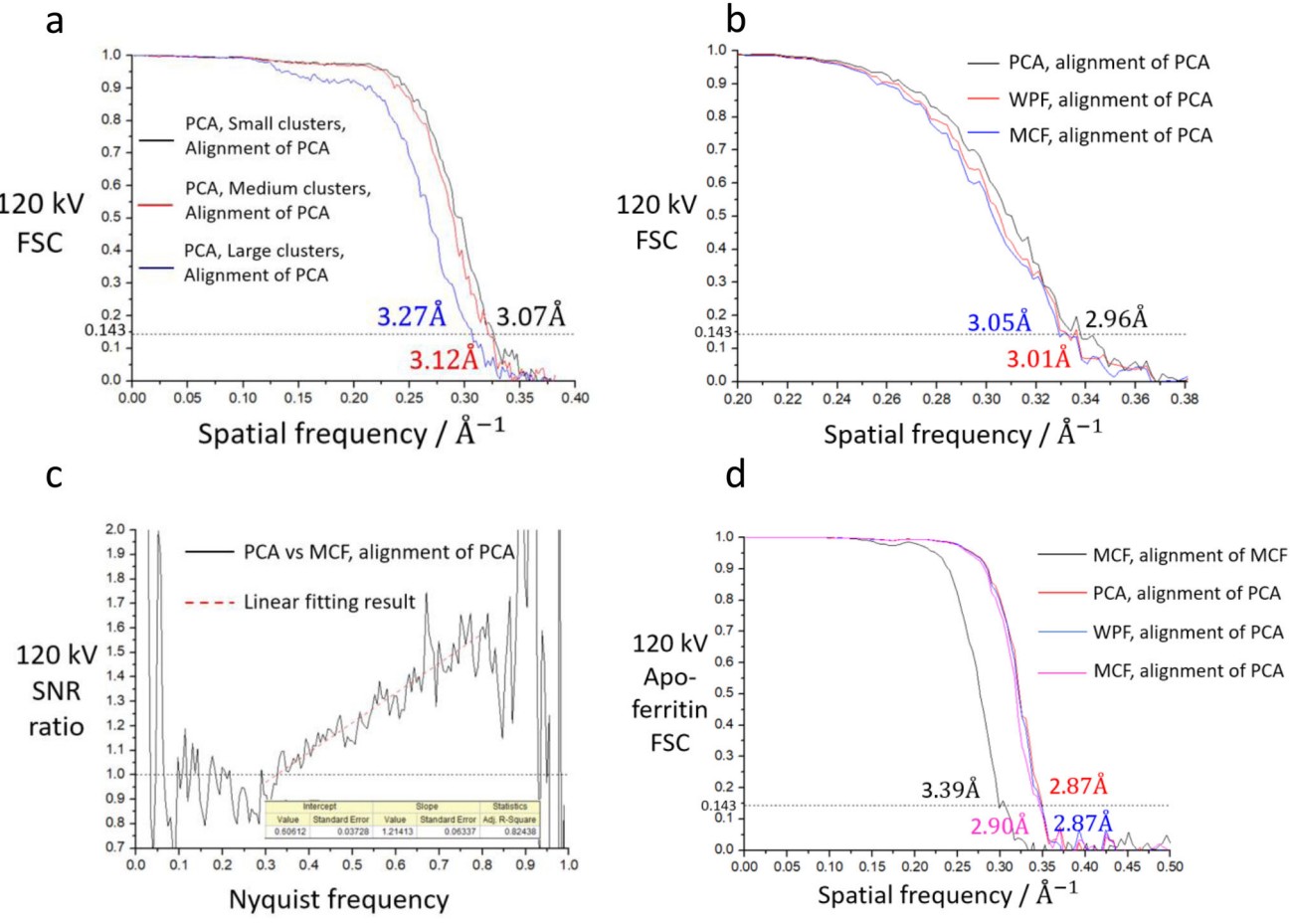

**Fig. 3 Performances of PCA and WPF at 120 kV. a** FSC curves for 3 types of clusters in Coxsackievirus A10 dataset at 120 kV with PCA counting algorithm. Each pair of reconstructions holds roughly the same number of clusters (~81.4% of 'small-sized' clusters, ~71.3% of 'medium-sized' clusters and 100% of 'large-sized' clusters). **b** FSC curves of PCA, WPF and MCF counting algorithm in Coxsackievirus A10 dataset at 120 kV. All clusters are used. **c** SNR ratio between PCA and MCF. The linear fitting interval is 0.30~0.80 Nyquist frequency. **d** FSC curves of PCA, WPF and MCF counting algorithm in apo-ferritin dataset at 120 kV. Alignment of MCF means we re-do SPA process since CTF estimation, and particles are re-picked, re-classified and re-refined within images from MCF algorithm. All the resolution of reconstructions based on gold-standard FSC criterion marks on the graph.

**Key elements of the 120-kV high-end cryo-TEM**. The SNR variance for large-sized clusters obtained from the MCF and PVF methods indicates that the peak of the cluster represents the actual position of the incident electron better than the centre-of-mass position. For large-sized clusters, a background signal that is weak comparing to peak but strong enough dictates the centre-of-mass position, which deviates from the peak signal produced by the incident electron. It is possible that the background signal is composed of back-scattered electrons (BSEs). To find the contribution of BSEs to the DDD, we conducted Monte-Carlo simulations using the CASINO package[37] to calculate the BSE coefficient for a cuboid of pure silicon and the results shown in Supplementary Fig. 7a (Method).

At a voltage of 120 kV and thickness of 40 μm, the BSE rate is approximately 15%, which is five times higher than that at 300 kV. In addition, our simulations show that within a radius of 20 μm, majority of the BSEs leave the same surface of silicon from which the primary electron enters at 120 kV (Supplementary Fig. 7b). That is, the BSEs may produce a background signal with a size similar to that of large-sized clusters. To reduce the production of large-sized clusters, if the BSE rate at 120 kV is similar as that at 300 kV, our simulations show that the camera chip should be back-thinned to 10 μm, which is much thinner than current thicknesses of 30 μm of Falcon III.

The final resolutions of Coxsackievirus A10 and apo-ferritin are lower than what they should be at 300 kV, which we believe arises from chromatic aberration. The depressed FSC' and corresponding SNR' by chromatic aberration at lower voltage can be estimated by FSC obtained at 300 kV (Method and Supplementary Note 9). To test the chromatic aberration depression directly via various voltages, we acquired Coxsackievirus and apo-ferritin datasets at 300 kV with the Titan Krios microscope and K2 camera separately and made a 3D-reconstruction with the same number of particles as at 120 kV. We compared the depressed FSC' with the actual FSC obtained at 120 kV (Fig. 4c, d). Clearly, the chromatic aberration was the main limitation, and the resolution of the 3D-reconstruction at 120 kV would be no worse than that at 300 kV if the chromatic aberration envelope depression effect was eliminated.

## Discussion

Our measurement of critical dose at 300 kV is systematically smaller than the results of Grant's[17]. A deviation of gain factor in K3 may be a contributor. We scaled our absolute critical dose in order to fit their measurements at 300 kV (Fig. 1c). The relative ratio had not been affected. B-factor plots from RELION polish provides a qualitative view of radiation damage (Supplementary Fig. 8), which also indicated the damage per-electron became severer as accelerate voltage decreased.

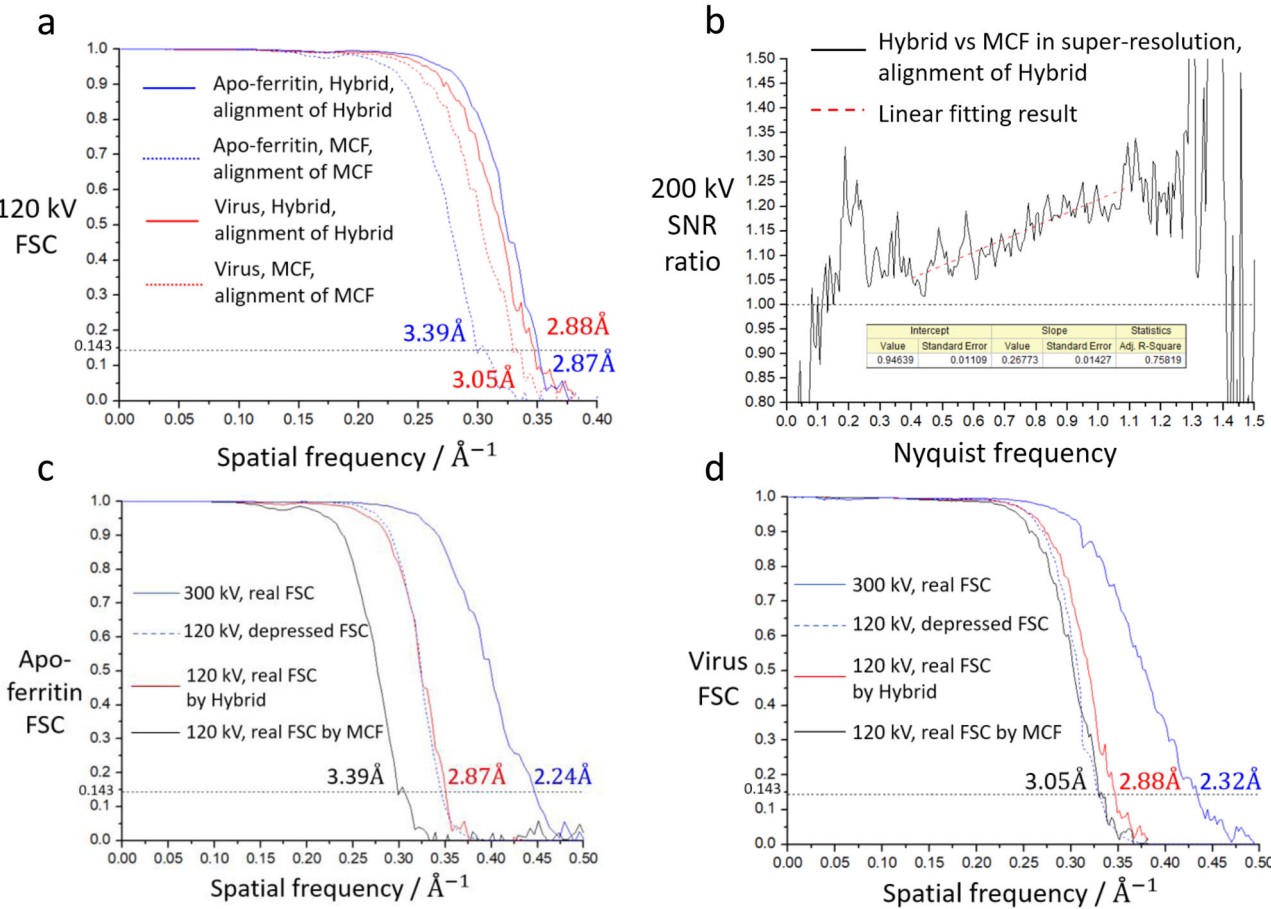

**Fig. 4 Performances of Hybrid algorithm and low-voltage cryo-EM. a** FSC curves of apo-ferritin and Coxsackievirus A10 dataset at 120 kV with Hybrid and MCF counting method. **b** SNR ratio between Hybrid and MCF at 200 kV, both in super-resolution mode. The linear fitting interval is 0.40~1.10 Nyquist frequency. **c** Predicted depressed FSC and reality FSC curves of apo-ferritin. Depressed FSC is calculated by Eqs. 4 to 6. **d** Predicted depressed FSC and reality FSC curves of Coxsackievirus A10. Depressed FSC is calculated by Eqs. 4 to 6. All the resolution of reconstructions based on gold-standard FSC criterion marks on the graph.

To estimate the total signal produced by the incident electrons before proteins are completely damaged, Peet et al.[15] suggested an IFC of the form

$$IFC = T \frac{\sigma_e}{\sigma_i}, \quad (1)$$

where $T$ denotes the total transmission of electron representing single-scattered electrons; $\sigma_e$ and $\sigma_i$ denote the elastic and inelastic scattering sections, representing signal and radiation damage. They used $\sigma_e$ from Langmore and Smith[13], and $\sigma_i$ from the Estar database to calculate IFC. We only measured vitrified protein with $T \sim 30$ nm, meaning our prediction of IFC being valid only for this thickness. Furthermore, a significant deviation in the empirical function and Estar database at lower voltages implies their prediction of IFC will be diverse. We suggest more precise measurements to be performed to the components of IFC from vitrified samples below 100 kV.

An interesting finding is that $pow = 0$ is better than $pow = 1$ for large-sized clusters with the PCA. When $pow = 0$, all pixels inside the cluster have the same possibility to represent the position of the incident electron. This prompts the notion that the shape-related spreading function or the corresponding envelope in Fourier space is the critical factor that improves the SNR by compressing noise. We examined this assumption by calculating the modulation transfer function and NPS curves for $pow = 0$ and $pow = 1$ for the large-sized clusters (Supplementary Fig. 7c) which suggests that $pow = 0$ does not provide an extra signal while suppressing noise.

We have shown that signal depression by chromatic aberration is a major limiting factor in low-voltage cryo-EM. For example, the SNR at frequency of $1/5$ Å$^{-1}$ drops ~35% in a 120-kV TEM compared with a 300-kV TEM (Supplementary Fig. 9a, chromatic aberration coefficient ($Cc$) = 2.7 mm, $\delta E$ = 0.7 eV). Several envelope functions with changes in parameter values for $Cc$, $\delta E$, and voltage (Supplementary Fig. 9a–c) enable us to infer that for 100~120-kV cryo-EM, $\delta E$ = 0.3 eV typical of a cold field-emission gun or a smaller $Cc$ value such as $Cc$ = 1.0 mm is necessary when a similar performance as at 300 kV ($Cc$ = 2.7 mm and $\delta E$ = 0.7 eV) is required. When the voltage decreases to 80 kV, $\delta E$ = 0.1 eV typical of a monochromatic device or $Cc$ = 0.5 mm is required. The size of Streptavidin, a 52 kD protein, is practically close to the resolving limit for 300-kV cryo-EM. Any degradation to the SNR of the image may prevent a successful reconstruction. We collected cryo-EM data of Streptavidin at both 120 kV and 300 kV in a Titan Krios. 2D-classification showed the decreasing of SNR at high frequency of 120-kV TEM resulted in misalignment (Supplementary Fig. 10), which agreed with the envelop function. Further 3D reconstruction of the 120-kV dataset also led to misalignment of images and cannot provide any detailed structure.

Our results in apo-ferritin dataset at 120 kV had already shown that an improvement in SNR of cryo-EM images by our counting

algorithm led to a better alignment, which improved the resolution of 0.52 Å. While for the 30 nm virus, the improvement of resolution is 0.17 Å, thus, the improved alignment is marginal. These results indicated that smaller proteins are benefited more from our counting algorithm. To further confirm this finding, we subtracted the symmetry-expanded 120-kV apo-ferritin dataset (Supplementary Note 10) with masks containing of different masses. After local refinement, more SNR improvement was obtained with less mass remaining (Supplementary Fig. 11).

## Methods

**Calibration of TEM for critical dose measurement**. To measure critical doses, electron doses to sample as well as the pixel size of the images need to be known. Because the pixel size of images from this microscope operating at 300 kV was carefully calibrated by measuring of the lattice of gold crystal, pixel sizes at voltages 160 kV and 120 kV were calculated by scaling the reconstruction from each dataset with the reconstruction from the 300-kV dataset. 'Fit in map' function in UCSF Chimera[38] was used to calculate the correlation coefficient (CC) value of rescaled map to model (from 300-kV reconstruction). The search range is 1.01 to 1.05 Å, with step size of 0.005 Å. Usually, an average dose for each pixel of the K3 camera can be calculated based on the overall counts for an image according to the manufacturer's calibrated ratio that converts counts to doses. However, the ratio is unknown when the microscope is operated at 160 kV and 120 kV. In addition, measuring the actual doses for a sample without a Faraday cup at the sample site is unfeasible. Therefore, a method is designed that choose the value of the current of the fluorescent screen shown as 'Screen Current' (SC) as a reference to measure relative doses under the assumption that the screen current being proportional to the number of electrons hitting the screen irrespective of energy[39,40]. In addition, the homogeneity of the electron beam on the screen was found good enough for measurements. The electron beam provided parallel illumination within an area of 6.3 μm to 12.5 μm. The beam was beam-shifted to centralise the five spots including four corners and the centre of the beam to the K3 detector (Gatan Inc.) (Supplementary Fig. 1a–c). The electron counts were recorded using the K3 detector and only vary from 0.66% to 0.69% (Supplementary Fig. 1d).

When the fluorescent screen is fully covered by the electron beam and the SC is measured, only a fraction of the electrons entering the K3 are detected. The fraction is correlated with the relative magnification between the screen and K3. The relative magnification may be different at 300 kV, 160 kV, and 120 kV. Therefore, a multiple-point-fitting method (Supplementary Note 2 and Supplementary Fig. 2) was developed to measure the relative magnifications at the corresponding acceleration voltages. The raw fitting results of the K3 readout of the SC at various acceleration voltages after relative magnification calibration are shown in Fig. 1b; the counting scaling factors (CSFs, Supplementary Note 3) representing the gain conversion of electrons are shown on Supplementary Table 1.

The dose rate for the sample at each dataset was calculated in accordance with CSFs and readouts of K3 detector. Reconstructions the 300-kV, 160-kV and 120-kV datasets were calculated (See parts below) and gave a final resolution of 3.11 Å, 3.46 Å, and 3.49 Å, respectively. In each dataset, the per-frame reconstructions were calculated[17]. At 300 kV/160 kV/120 kV, the critical doses were calibrated using a voltage dependent CSF to convert the readout of K3 to electrons (Supplementary Table 1 and Fig. 1c).

**Calculating clusters from raw Falcon III images**. A Laplace-of-Gaussian-like convolution kernel[41] was applied to the raw frame to generate a binary-value map. In this map, any pixel of the raw frame with its value above a given threshold was set to 1, otherwise set to 0 (Supplementary Note 5). A connected-component-labelling algorithm[42] was used to search for the nearest-neighbour pixels and to group these pixels into a cluster (Supplementary Note 5).

**Producing Image$_{small/medium/large}$ and measuring DQE of Falcon III detector**. The same number (~27% of total electrons) of the small, medium, and large-sized clusters for the 120-keV electrons were randomly selected in each frame and summed over all frames to produce three counting images, image$_{small}$, image$_{medium}$, and image$_{large}$, respectively. When applied to Coxsackievirus and apo-ferritin datasets obtained at 120 kV, for each counting algorithm, reconstructions of images$_{small}$, images$_{medium}$ and images$_{large}$ were calculated separately using the same alignment file generated by RELION (Supplementary Note 7). For testing *pow* parameter, *pow* = 0, ±0.25, ±0.5, ±0.75, ±1.0, ±1.25, ±1.5, ±2.0, ±3.0 were used to produce Image$_{large}$. *pow* = 0, ±0.5, ±1.0, ±2.0 were used to produce Image$_{small/medium/all}$. Since no comparison was need between all types, every cluster was used instead of part mention above. Only some of those FSC plots were shown for a clearer view and there was no effect to result. 'Knife-edge'[43] images containing 8850 frames was exposed for 225 s with a dose rate of 0.6 e-/s/pixel monitored by EPU software (Thermo Fisher Scientific) using the Falcon III camera. FINDDQE software[33] was used to measure the DQEs of the three images.

**SNR weighting of counting algorithm**. The SNR curve is obtained from the FSC curve of the reconstruction using the conversion

$$SNR = \frac{2\,FSC}{1 - FSC}, \qquad (2)$$

proposed by Grigorieff[44]. Taking an example in Supplementary Fig. 4d, at half the Nyquist frequency, the FSC values from the PVF and MCF methods are 0.80 and 0.68. Hence, the corresponding SNR is 8.00 and 4.25, respectively.

The large gap in the FSCs for the three types of clusters at 120 kV suggests that the SNR of the electrons should be summed based on its SNR weight. The weighting function by maximising the SNR of the summed image and applied the weight to the large-sized clusters (Supplementary Note 8) using

$$w = \sqrt{\frac{SNR_2}{SNR_1}} \cdot \sqrt{\frac{NPS_1}{NPS_2}}. \qquad (3)$$

To demonstrate, the ratio $\frac{SNR_2}{SNR_1}$ was calculated (Supplementary Note 7) for large-sized clusters obtained from the PVF method and small-sized clusters obtained with the MCF method at 120 kV (Supplementary Fig. 4g).

**Cryo-EM and data collection**. For cryo-grid preparation of critical dose measurement, a 3 μL PBS containing of Coxsackievirus A10 was applied to a fresh glow discharged 300-mesh holey carbon-coated copper grid (GIG, R2/1). Grids were blotted for 3 s twice with blot force being set to −0.2 and in 99% humidity by plunge-freezing device (CP3; Gatan Inc.) in liquid ethane at −183 °C.

For cryo-grid preparation for the testing of electron counting algorithm, a 3 μL PBS containing of Coxsackievirus A10 was applied to a fresh glow discharged 300-mesh holey carbon-coated copper grid (GIG, R2/1 + c) with super-thin layer of carbon (<3 nm). Grids were blotted for 3 s with blot force being set to −0.2 and in 99% humidity by plunge-freezing device (CP3; Gatan Inc.) in liquid ethane at −183 °C. A 3 μL PBS containing of apo-ferritin was applied to a fresh glow discharged 300-mesh holey Ni-Ti grid (R1.2/1.3). Grids were blotted for 5 s with blot force being set to 2 and in 99% humidity by plunge-freezing device (Vitrobot; Thermo Fisher Scientific) in liquid ethane at −183°C.

For critical dose measurements, cryo-EM data sets were collected with the same grid of Coxsackievirus A10 sample at 300 kV, 160 kV and 120 kV with a Titan Krios microscope (Thermo Fisher). Doses at 160 kV and 120 kV were calibrated by CSF. Movies (60 frames, each 0.058 s, total dose 60.49 e-/Å2 at 300 kV. 50 frames, each 0.1 s, total dose 53.92 e-/Å2 at 160 kV. 60 frames, each 0.1 s, total dose 36.82 e-/Å2 at 120 kV.) were recorded using a K3 detector in non-CDS, bin-0.5 mode with a defocus range of −0.6 to −1.8 μm. Automated single particle data acquisition was performed with SerialEM[45] at each voltage, and via beam-tilt data collection[46] at 300 kV, with a nominal magnification of 22,500×, which yields a final pixel size of 1.07 Å at 300 kV, 1.03 Å at 160 kV and 120 kV.

For electron counting, cryo-EM data sets were collected at 120 kV and 200 kV of Coxsackievirus A10 samples and at 120 kV and 300 kV of apo-ferritin samples with a Titan Krios microscope (Thermo Fisher). Movies (2331 frames, each 0.0256 s, total dose about 18.5 cluster/Å2 at 120 kV; 1943 frames, each 0.0256 s, total dose about 10.1 cluster/Å2 at 200 kV of Coxsackievirus A10. 1554 frames, each 0.0256 s, total dose about 19.6 cluster/Å2 at 120 kV; 1566 frames, each 0.0256 s, total dose about 29.1 cluster/Å2 at 300 kV of apo-ferritin.) were recorded using a Falcon III detector with a defocus range of −0.6 to −2.5 μm. SNR Tool software was used and MFCDS mode was triggered in order to acquire pre-electron-counting images. Pre-EC gain was acquired and applied. Automated single particle data acquisition was performed with TEM Imaging & Analysis software (TIA) and scripts. A nominal magnification of 59,000×, which yielded a final pixel size of 1.31 Å at 120 kV of Coxsackievirus A10 was used. A nominal magnification of 47,000×, which yielded a final pixel size of 1.70 Å at 200 kV of Coxsackievirus A10 was used. A nominal magnification of 75,000×, which yielded a final pixel size of 1.06 Å at 300 kV and 1.00 Å at 120 kV of apo-ferritin was used.

**Image processing for critical dose measurement**. The movie stacks collected by K3 were frame-shift corrected and binned 2x by MotionCor2[47]. Defocus values of micrographs were determined by CTFFIND4[48] to be in the range of −0.5 to −3.5 μm. A total of 20,294 particles at 300 kV, 21,137 particles at 160 kV, 25,237 particles at 120 kV (448 by 448 pixels for all voltages) were used from 555, 655 and 697 selected micrographs at 300 kV, 160 kV and 120 kV, respectively, for final reconstructions. Procession of the three data sets was performed with RELION3[49] with I3-symmetry imposed. After convergence of auto-refinement for three datasets, CTF refinement and Bayesian Polish[50] were performed by RELION3.1. After convergence of final refinement for both halves, the resolution was determined to be 3.11 Å at 300 kV, 3.46 Å at 160 kV and 3.49 Å at 120 kV on the basis of the gold-standard FSC = 0.143 criterion.

**Image processing for electron counting tests**. To reduce disk usage, the image frames after electron counting were simple summed. Number of frames to be summed up was controlled by a parameter '−p' in our counting programmes. The Coxsackievirus A10 dataset at 120 kV and apo-ferritin dataset at 300 kV used '−p 40', meaning simple sum of every 40 frames, the Coxsackievirus A10 dataset at 200 kV used '−p 45', and apo-ferritin dataset at 120 kV used '−p 30'. Those stacks

**Tables 1 Cryo-EM data collection, refinement and validation statistics.**

| | #1 Structure of Coxsackievirus A10 for critical dose measurement at 120 kV (EMD-32600) | #2 Structure of Coxsackievirus A10 for critical dose measurement at 160 kV (EMD-32601) | #3 Structure of Coxsackievirus A10 for critical dose measurement at 300 kV (EMD-32602) |
|---|---|---|---|
| Data collection and processing | | | |
| Magnification | 22,500x | 22,500x | 22,500x |
| Voltage (kV) | 120 | 160 | 300 |
| Electron exposure (e–/Å²) | 36.82 | 53.92 | 60.49 |
| Defocus range (μm) | −0.6-−1.8 | −0.6-−1.8 | −0.6-−1.8 |
| Pixel size (Å) | 1.03 | 1.03 | 1.07 |
| Symmetry imposed | I3 | I3 | I3 |
| Initial particle images (no.) | 70,125 | 50,715 | 35,755 |
| Final particle images (no.) | 25,237 | 21,137 | 20,294 |
| Map resolution (Å) FSC threshold | 3.49 | 3.46 | 3.11 |
| Map resolution range (Å) | >3.49 | >3.46 | |
| Refinement | | | |
| Map sharpening $B$ factor (Å²) | −223 | −212 | −176 |
| | #4 Structure of Coxsackievirus A10 with Hybrid electron counting at 120 kV (EMD-32603) | #5 Structure of Coxsackievirus A10 with MCF electron counting at 120 kV (EMD-32604) | #6 Structure of Coxsackievirus A10 with WPF electron counting at 120 kV (EMD-32605) |
| Data collection and processing | | | |
| Magnification | 59,000x | 59,000x | 59,000x |
| Voltage (kV) | 120 | 120 | 120 |
| Electron exposure (e–/Å²) | 18.5 | 18.5 | 18.5 |
| Defocus range (μm) | −0.6-−2.5 | −0.6-−2.5 | −0.6-−2.5 |
| Pixel size (Å) | 1.31 | 1.31 | 1.31 |
| Symmetry imposed | I3 | I3 | I3 |
| Initial particle images (no.) | 50,101 | 50,101 | 50,101 |
| Final particle images (no.) | 10,517 | 10,517 | 10,517 |
| Map resolution (Å) FSC threshold | 2.88 | 3.05 | 2.97 |
| Map resolution range (Å) | >2.88 | >3.05 | >2.97 |
| Refinement | | | |
| Map sharpening $B$ factor (Å²) | −159 | −150 | −159 |
| | #7 Structure of Coxsackievirus A10 with MCF electron counting and large-sized clusters at 120 kV (EMD-32606) | #8 Structure of Coxsackievirus A10 with PVF electron counting and large-sized clusters at 120 kV (EMD-32607) | #9 Structure of Coxsackievirus A10 with PCA electron counting and large-sized clusters at 120 kV (EMD-32608) |
| Data collection and processing | | | |
| Magnification | 59,000x | 59,000x | 59,000x |
| Voltage (kV) | 120 | 120 | 120 |
| Electron exposure (e–/Å²) | 18.5 | 18.5 | 18.5 |
| Defocus range (μm) | −0.6-−2.5 | −0.6-−2.5 | −0.6-−2.5 |
| Pixel size (Å) | 1.31 | 1.31 | 1.31 |
| Symmetry imposed | I3 | I3 | I3 |
| Initial particle images (no.) | 50,101 | 50,101 | 50,101 |
| Final particle images (no.) | 10,517 | 10,517 | 10,517 |
| Map resolution (Å) FSC threshold | 4.35 | 3.66 | 3.27 |
| Map resolution range (Å) | >4.35 | >3.66 | >3.27 |
| Refinement | | | |

**Table 1 (continued)**

| | | | |
|---|---|---|---|
| Map sharpening $B$ factor (Å$^2$) | −220 | −185 | −179 |
| | **#10 Structure of Coxsackievirus A10 with MCF electron counting at 200 kV (EMD-32609)** | **#11 Structure of Coxsackievirus A10 with Hybrid electron counting at 200 kV (EMD-32611)** | |
| Data collection and processing | | | |
| Magnification | 47,000x | 47,000x | |
| Voltage (kV) | 200 | 200 | |
| Electron exposure (e−/Å$^2$) | 10.1 | 10.1 | |
| Defocus range (µm) | −0.6-−2.5 | −0.6-−2.5 | |
| Pixel size (Å) | 0.85 after 2 × 2 super-resolution | 0.85 after 2 × 2 super-resolution | |
| Symmetry imposed | I3 | I3 | |
| Initial particle images (no.) | 62,833 | 62,833 | |
| Final particle images (no.) | 11,048 | 11,048 | |
| Map resolution (Å) FSC threshold | 2.72 | 2.66 | |
| Map resolution range (Å) Refinement | >2.72 | >2.66 | |
| Map sharpening $B$ factor (Å$^2$) | −159 | −157 | |
| | **#12 Structure of apo-ferritin with PCA electron counting at 300 kV (EMD-32610)** | **#13 Structure of apo-ferritin with Hybrid electron counting at 120 kV (EMD-32612)** | **#14 Structure of apo-ferritin with MCF electron counting at 120 kV (EMD-32613)** |
| Data collection and processing | | | |
| Magnification | 75,000x | 75,000x | 75,000x |
| Voltage (kV) | 300 | 120 | 120 |
| Electron exposure (e−/Å$^2$) | 29.1 | 19.6 | 19.5 |
| Defocus range (µm) | −0.6-−2.5 | −0.6-−2.5 | −0.6-−2.5 |
| Pixel size (Å) | 1.06 | 1.00 | 1.00 |
| Symmetry imposed | O | O | O |
| Initial particle images (no.) | 52,692 | 150,471 | 150,471 |
| Final particle images (no.) | 25,930 | 88,485 | 54,088 |
| Map resolution (Å) FSC threshold | 2.51 | 2.87 | 3.39 |
| Map resolution range (Å) Refinement | >2.51 | >2.87 | >3.39 |
| Map sharpening $B$ factor (Å$^2$) | −91 | −169 | −152 |

were then frame-shift corrected by MotionCor2. When MCF was used, the stacks were firstly binned into half by relion_image_handler programme then shift corrected by MotionCor2. Defocus values of micrographs were determined by CTFFIND4 to be in the range of −0.5 to −3.5 µm. A total of 10,517 and 11,048 particles of Coxsackievirus A10 at 120 kV (352 by 352 pixels with pixel size of 1.31 Å) and 200 kV (320 by 320 pixels with pixel size of 1.7 Å in non-super-resolution mode and 640 by 640 with pixel size of 0.85 Å in 2 × 2 super-resolution mode), 88,485 particles and 25,930 of apo-ferritin at 120 kV (224 by 224 pixels with pixel size of 1.00 Å) and 300 kV (192 by 192 pixels with pixel size of 1.06 Å) were used from 143, 141, 283 and 128 selected micrographs, respectively, for final reconstructions. Procession of the three data sets was performed with RELION3. Coxsackievirus imposed with I3-symmetry, and apo-ferritin imposed with O-symmetry. After convergence of auto-refinement for all datasets, CTF refinement and Bayesian Polish were performed by RELION3, then another round of auto-refinement was proceeded until convergence. The alignments of each filter motioned above that produced highest resolution were kept for SNR comparison. FSC curve was calculated and the resolution was determined to be 2.48 Å at 300 kV and 2.87 Å at 120 kV for apo-ferritin, 2.66 Å with super-resolution at 200 kV and 2.88 Å at 120 kV for Coxsackievirus A10 on the basis of the gold-standard FSC = 0.143 criterion. Density samples of residue of apo-ferritin at 120 kV with Hybrid and MCF counting method were shown in Supplementary Fig. 12. Cryo-EM data collection tables are shown in Table 1.

**Monte-Carlo simulations for back-scattering electrons**. With help of CASINO package, a pure silicon box of 1400 µm × 1400 µm in length/width (equivalently, 100 × 100 pixels with a pixel size of 14 µm) was set. 10,000 electrons were used in every run and initial impact position laid on the centre of upper surface. For each group of voltage/thickness, five times of simulations were performed.

**Estimating depressed FSC curves by chromatic aberration**. The envelope of chromatic aberration can be calculated using[28]:

$$E(u) = \exp[-0.5 \cdot (\pi\lambda\delta)^2 \cdot u^4], \tag{4}$$

$$\delta = Cc \cdot \sqrt{4\left(\frac{\Delta I}{I}\right)^2 + \left(\frac{\Delta E}{V}\right)^2 + \left(\frac{\Delta V}{V}\right)^2}. \tag{5}$$

The parameters used for $\frac{\triangle I}{I}$ and $\frac{\triangle V}{V}$ came from Nakane et al.[51] After $E(u)$ is obtained, we calculate the depressed SNR' from

$$SNR' = E'(u')^2 \cdot \frac{SNR}{E(u)^2} \qquad (6)$$

The depressed FSC' then can be calculated by depressed SNR' (Supplementary Note 9).

**Reporting summary**. Further information on research design is available in the Nature Research Reporting Summary linked to this article.

## Data availability

The EM density maps have been deposited in the Electron Microscopy Data Bank under accession codes EMD-32603/32604/32605 (Hybrid, MCF and WPF counting of Coxsackievirus A10 at 120 kV), EMD-32606/32607/32608 (MCF, PVF and PCA counting of large-sized clusters of Coxsackievirus A10 at 120 kV), EMD-32609/32611 (MCF, Hybrid counting of Coxsackievirus A10 at 200 kV), EMD-32600/32601/32602 (120/160/300 kV Coxsackievirus A10 from critical dose measurement), EMD-32610 (PCA counting of apo-ferritin at 300 kV), EMD- 32612/32613 (Hybrid, MCF counting of apo-ferritin at 120 kV). Source data files are provided in Supplementary Data 1. Other data that support the findings of this study are available from the authors upon request.

## Code availability

The binary builds of electron counting programmes are available at https://github.com/homurachan/Electron_Counting. Source codes of modified version of RELION are available at https://github.com/homurachan/Modified_RELION3. Other source codes or scripts used are available at https://github.com/homurachan/Misc_Scripts.

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

## Acknowledgements

We thank Dr. L. Jin at Guangzhou Regenerative Medicine and Health Guangdong Laboratory, Guangzhou, China for discussing DDD and counting algorithm. We thank Dr. S. Liu at the Institute of Microbiology, Chinese Academy Sciences, Beijing, China for providing Coxsackievirus A10 protein. We thank Dr. K.L. Fan at the Institute of Biophysics, Chinese Academy Sciences, Beijing, China for providing apo-ferritin protein. We thank Dr. H.W. Wang and Dr. C. Liu at School of Life Sciences, Tsinghua University, Beijing, China for providing the cryo-EM samples of apo-streptavidin. We thank L.F. Kong at the Institute of Biophysics, Chinese Academy Sciences, Beijing, China for cryo-EM data management. Cryo-EM data collection was carried out at the Centre for Biological Imaging, Core Facilities for Protein Science at the Institute of Biophysics (IBP), Chinese Academy of Sciences (CAS). We thank X.J. Huang, B.L. Zhu, X.J. Li, L.H. Chen, F. Sun, and other staff members at the Centre for Biological Imaging (IBP, CAS). The project was funded by the National Natural Science Foundation of China (31930069 and 32150010), the National Key R&D Programme of China (2017YFA0504700 and 2021YFA1301501), the Strategic Priority Research Programme of the Chinese Academy of Sciences (XDB37040101) and the Key Research Programme of Frontier Sciences at the Chinese Academy of Sciences (ZDBS-LY-SM003). The project was supported by the Scientific Instrument Developing Project of the Chinese Academy of Sciences, Grant No.ZDKYYQ20170002.

## Author contributions

X.Z.Z. conceived the project; D.J.Z. provided or modified all codes; X.Z.Z. and D.J.Z. designed the experiments; H.G.S. and C.L.W. prepared the cryo-EM samples; D.J.Z. conducted the EM analysis; H.G.S., C.L.W., and D.J.Z. performed the cryo-EM data collection; D.J.Z. processed the cryo-EM data and reconstructed the cryo-EM map; X.Z.Z. and D.J.Z. wrote the manuscript; all authors discussed and commented on the results and the manuscript.

## Competing interests

The authors declare the following competing interests: A patent application on electron cluster classification and electron counting algorithm of WPF/Hybrid is pending.
