## [Peer Review File · Communications Biology]

Reviewers' comments:

Reviewer #1 (Remarks to the Author):

The manuscript by Zhu et al describes a new counting algorithm that can be used at lower voltage(s) with existing direct electron detectors. There is now a lot of traction to use lower voltage for many biological imaging in particular for single particle cryoEM. Apart from technical reasons, the cost of a 300 kV microscope and its maintenance is another prime factor in looking for affordable TEMs. An article in 2019 from Naydenova et al had shown practical demonstration of high-resolution structures at 100 keV with a hybrid pixel detector. An important point to consider is that the detector pixel size was large and the area was small, which means more EM time. The design and optimization of detectors at different voltages is an ongoing project by different groups and to overcome this in the interim period, the authors in this manuscript take the existing direct detector and perform electron counting with a new algorithm. Though, these detectors will have lower DQE at lower voltage, the results in the manuscript show that it is possible to obtain high-resolution reconstructions limited by the chromatic aberration. It is a timely contribution to the use of low-voltage microscopy for cryoTEM.

The manuscript can be improved with more clear writing and coherence for a general audience to read. In particular for all the new cryoEM people who are now starting to pursue single particle EM. At the moment, there are lot of terms and figures, which are difficult to follow. Below are suggestions/comments that I believe will make the manuscript read better.

- 1) Abstract – this can be made easier to read. Line 13 – should read as ‘One of the major barriers’ rather than calling it the one barrier
- 2) Line 19 – sentence needs rephrasing
- 3) The first paragraph of introduction has no references cited. It will be useful to include some such as Merk et al 2016; Cheng, Y 2018; Cheng et al 2015 (not necessarily these but along these lines). Also, a reference to radiation damage is also essential (first sentence).
- 4) Line 46 – please change unattainable to difficult. The use of ‘unattainable’ seems to indicate impossible but as the field is progressing fast, this could be a strong statement.
- 5) Line 68 – ‘(HPC) is gaining’ please change to (HPC) has a higher DQE for lower energy electrons
- 6) Line 93-95 – the sentence referring chromatic aberration is unclear and needs rephrasing.
- 7) How the pixel sizes were calibrated for each voltage needs to be mentioned in a bit more detail (perhaps in methods) and why will there be a difference in pixel size for same nominal magnification at different voltage can also be mentioned in a sentence (for general readers)
- 8) Will it be possible to show the B-factor plots (from relion polishing) for coxsackievirus A at different voltages. Perhaps as supplementary figure.
- 9) Line 135-138 – the sentences are bit confusing. Please rephrase
- 10) Line 174, line 200 – there are abbreviations but these have not been mentioned before such as MC, PEEK, GC. Only in supplementary material these are defined. Please expand the abbreviations when mentioning these in the first instance – Peak value filter (PEEK) and Geometric centre filter (GC).
- 11) Line 183 – ‘using the same alignment file’, where did this alignment file come from. Please clarify
- 12) Line 282/288 – instead of ‘push’ or ‘pushed’ – change to increase and increased
- 13) Line 334 – BSE should be expanded to Back-scattered electrons
- 14) Line 400,464 – Is CP3 not from Gatan?
- 15) Line 400,405,407 – ‘liquid ethane on -183°C’ to liquid ethane at -183°C
- 16) Line 403 – it is mentioned that the grids coated with super thin layer of carbon. Was it 2Å or 5 Å or 10Å?
- 17) Line 406 – why was Ni-Ti used for this. Any particular reason?
- 18) Line 410 – ‘doses at 160/120, calibrated by CSF’ What is CSF? (it is mentioned in the supplement but useful to expand it here as counting scaling factor)
- 19) Line 426 – TIA was used for automatic data acquisition. This is surprising, how was it done.
- 20) Line 430 – it is mentioned that 96,000x @ 300 kV corresponds to 1.06 Å and 1 Å at 120 kV. But, if I remember right, the pixel size of Krios at 96,000 x is around 0.85 Å. Please check these. (Also, all the pixel values at 120 kV are lower than at 300 kV and only this is larger). In line 454, it is mentioned that 0.85 Å was used but not sure if this was apoferritin data mentioned in line 430.

21) Would it be possible to mention the gun lens (and or extraction voltage) used for different voltages. I guess the gun lens was higher at lower voltage.

22) Figure legend 1 – line 606, correct the sentence to 'image was taken'

Supplementary material:

Supplementary note 2: line 313 – change to 'data was acquired at a magnification ..'

In supplementary note 6 – in several places (line 400,401,413) formula is mentioned but it is not clear which formula it is being referred to. It just says supplementary 3 or 5 or 6. But there is none in supplementary 5. I think it should Eq.2, Eq.5 and so on.

Line 411, please change implied to implemented

Reviewer #2 (Remarks to the Author):

The manuscript by Zhu et al. propose a new counting algorithm for lower electron energies (200 keV and 120 keV) that is claimed to be able to surpass the performance of the most widely used 300 keV instrumentation, particularly for small proteins (<100 kD in molecular mass). Despite the technical breakthroughs in hardware and software for biological cryo-EM studies, structural determination of small biomolecules remains very challenging as the authors rightly pointed out. However, the manuscript is quite challenging for non-experts with too many technical jargons that are not well explained. Most importantly, the authors failed to present a real biological case of which the molecular mass is sufficiently small to demonstrate the advantage of the proposed algorithm. Instead, the test cases are big with high symmetries. It is not trivial to evaluate how the proposed algorithm will actually overcome the size limitations in a real world. This reviewer therefore cannot recommend the publication of this manuscript in Communications Biology. Specific comments are as follows:

1. The advantage of using a lower electron energy has been discussed by Chris Russo and colleagues in 2019 (ref. 15) in which a wide range of molecular masses was tested to demonstrate the advantages of collecting data at 100 keV. While this manuscript provides a great deal of technical details by benchmarking against different instrumentation setups and data processing algorithms, it appears to miss the most important point that such a development should indeed help improve the data quality of small protein systems. The authors should use some model systems that have been evaluated before, such as haemoglobin (64 kD), to demonstrate the improved performance.
2. There are several technical terms that require better definitions and explanations. For example, the terms PEEK and GC (line 200) were not defined throughout the manuscript.
3. Even for the selected model systems that are large and highly symmetric, the authors did not provide the actual cryo-EM maps derived from different algorithms together with the atomic models, making it difficult to judge the performances simply based on the FSC curves.
4. As a minor comment, many plots have very small labels, which limited the readability.

Reviewer #3 (Remarks to the Author):

The authors present a very welcome new electron counting algorithm for CryoEM, especially well-suited to 100 and 200 kV. The algorithm is sound and its evaluation very thorough. The article is generally well written and there are only a few aspects that could be improved:

- One wonders if the parameter pow can be considered a continuous variable, instead of discrete (0, 1, 2, ...) and better results would be obtained.

- Suppl. Table 1. It would be good if the input data is shown and the R2 of the regression reported.

- It is unclear why they needed to modify Relion3. Now that Relion4 is out, how portable are the changes?
- The EMDB maps have no public code.

Reviewer #1 (Remarks to the Author):

The manuscript by Zhu et al describes a new counting algorithm that can be used at lower voltage(s) with existing direct electron detectors. There is now a lot of traction to use lower voltage for many biological imaging in particular for single particle cryoEM. Apart from technical reasons, the cost of a 300 kV microscope and its maintenance is another prime factor in looking for affordable TEMs. An article in 2019 from Naydenova et al had shown practical demonstration of high-resolution structures at 100 keV with a hybrid pixel detector. An important point to consider is that the detector pixel size was large and the area was small, which means more EM time. The design and optimization of detectors at different voltages is an ongoing project by different groups and to overcome this in the interim period, the authors in this manuscript take the existing direct detector and perform electron counting with a new algorithm. Though, these detectors will have lower DQE at lower voltage, the results in the manuscript show that it is possible to obtain high-resolution reconstructions limited by the chromatic aberration. It is a timely contribution to the use of low-voltage microscopy for cryoTEM.

We thank the reviewer for the comments.

The manuscript can be improved with more clear writing and coherence for a general audience to read. In particular for all the new cryoEM people who are now starting to pursue single particle EM. At the moment, there are lot of terms and

figures, which are difficult to follow. Below are suggestions/comments that I believe will make the manuscript read better.

We thank the reviewer for the suggestions.

1) Abstract – this can be made easier to read. Line 13 – should read as ‘One of the major barriers’ rather than calling it the one barrier

We thank the reviewer for the suggestion and we have made the change.

2) Line 19 – sentence needs rephrasing

We thank the reviewer for the suggestion and we have modified the sentence to

“Here, we present an electron counting algorithm to improve the detection of low-energy electrons. The counting algorithm increased the SNR of 120-kV and 200-kV cryo-EM image from a Falcon III camera by 8%, 20% at half the Nyquist frequency and 21%, 80% at Nyquist frequency, respectively, resulting in a considerable improvement in resolution of 3D reconstructions.”

3) The first paragraph of introduction has no references cited. It will be useful to include some such as Merk et al 2016; Cheng, Y 2018; Cheng et al 2015 (not necessarily these but along these lines). Also, a reference to radiation damage is also essential (first sentence).

We thank the reviewer for the suggestions. We have cited mentioned papers as reference 5,6,7. In addition, four more papers have been cited as reference 1-4 about radiation damage.

4) Line 46 – please change unattainable to difficult. The use of ‘unattainable’

seems to indicate impossible but as the field is progressing fast, this could be a strong statement.

We thank the reviewer for the suggestion and have changed the unattainable to difficult.

5) Line 68 – '(HPC) is gaining' please change to (HPC) has a higher DQE for lower energy electrons

We thank the reviewer for the suggestion and we have made the change.

6) Line 93-95 – the sentence referring chromatic aberration is unclear and needs rephrasing.

We thank the reviewer for the suggestion and we have modified the sentence to "By estimating signal suppression being introduced from chromatic aberration, our results indicate that chromatic aberration is a major factor that limits the resolution of the 3D reconstruction from 120-kV cryo-EM."

7) How the pixel sizes were calibrated for each voltage needs to be mentioned in a bit more detail (perhaps in methods) and why will there be a difference in pixel size for same nominal magnification at different voltage can also be mentioned in a sentence (for general readers)

We used "fit in map" function in UCSF Chimera to calculate the correlation coefficient (CC) value of rescaled map to model (from 300-kV reconstruction). The search range is 1.01 to 1.05Å, with step size of 0.005Å. In the previous manuscript, CC searches were based on reconstructions of 3D-classification. The result in the table below is based on reconstruction after final refinement and post-process.

Pixel size of 160 kV / Å	CC (160 kV)	Pixel size of 120 kV / Å	CC (120 kV)
1.05	Map rotated	1.05	Map rotated
1.045	0.5543	1.045	0.3941
1.04	0.8172	1.04	0.7103
1.035	0.9481	1.035	0.901
1.03	0.9763	1.03	0.9734
1.025	0.9112	1.025	0.9505
1.02	0.7216	1.02	0.8203
1.015	0.3797	1.015	0.5399
1.01	0.2974	1.01	0.3058

We have corrected the pixel size of 160/120-kV at 22,500x with K3 from 1.02 Å to 1.03 Å. As a result, Plots in Figure 1c and 1d, total doses in Method were changed slightly.

However, the exact reason that difference in pixel size for same nominal magnification at different voltage is unknown to us. We have found that the same level of difference in pixel size existed among different 300kV microscopes at the same nominal magnification. Thus, the difference may come from the calibration of the microscopes.

8) Will it be possible to show the B-factor plots (from relion polishing) for coxsackievirus A at different voltages. Perhaps as supplementary figure.

We thank the reviewer for the suggestion, and the B-factor plots from RELION polish are shown below. We have added this figure to Supplementary Figure 8, and a sentence of " B-factor plots from RELION polish provides a qualitative view of radiation damage (Supplementary Figure 8), which also indicated the damage per-electron became severer as accelerate voltage decreased." in the discussion.

9) Line 135-138 – the sentences are bit confusing. Please rephrase

We thank the reviewer for the suggestion and have rephrased the sentence to "Comparing our result with that of Peet et al., we show that with decreasing acceleration voltage, the radiation damage per-electron is severer in the vitrified protein samples than in crystalized bacteriorhodopsin and paraffin. Our result is close to either the data from Estar database or an empirical function by Wall' s et al. (Fig. 1d). "

10) Line 174, line 200 – there are abbreviations but these have not been mentioned before such as MC, PEEK, GC. Only in supplementary material these are

defined. Please expand the abbreviations when mentioning these in the first instance – Peak value filter (PEEK) and Geometric centre filter (GC).

We thank the review for the comments. We have expanded the first abbreviations and re-named PEEK to PV for better understanding to readers.

11) Line 183 – ‘using the same alignment file’ , where did this alignment file come from. Please clarify.

We have modified the sentence and after to “For each image in the Coxsackievirus A10 dataset at 120 kV, 7 images with cluster sizes varying from 1 to 7 pixels and containing roughly the same number of randomly selected electrons were produced using the MC, which made 7 datasets to yield 7 reconstructions using the same alignment file generated by RELION, from the highest resolution reconstruction achieved latter in this work, with images produced by our improved counting algorithm.”

12) Line 282/288 – instead of ‘push’ or ‘pushed’ – change to increase and increased

We thank the reviewer for the suggestion and have changed the words.

13) Line 334 – BSE should be expanded to Back-scattered electrons

We thank the reviewer for the suggestion and have expanded the abbreviation.

14) Line 400,464 – Is CP3 not from Gatan?

We thank the reviewer for pointing out the error and we have corrected them.

15) Line 400,405,407 – ‘liquid ethane on -183 · C’ to liquid ethane at -183 · C

We thank the reviewer for pointing out the error and we have corrected it.

16) Line 403 – it is mentioned that the grids coated with super thin layer of carbon.

Was it 2Å or 5 Å or 10Å?

We used GIG 300 mesh R2/1+c grid. The factory claimed the thickness of layer of carbon was less than 3nm.

17) Line 406 – why was Ni-Ti used for this. Any particular reason?

The Ni-Ti grid means Ni-Ti film on gold grid. We found that our apo-ferritin achieves a better dispersion on the Ni-Ti grid as shown in figure below. Figure A shows ferritin on Ni-Ti 1.2/1.3 grid, and B shows ferritin on Quantifoil 1.2/1.3 grid.

This kind of grid is also cheaper than gold grid with gold film.

18) Line 410 – ‘doses at 160/120, calibrated by CSF’ What is CSF? (it is mentioned in the supplement but useful to expand it here as counting scaling factor)

We thank the reviewer for the suggestion and have expanded the abbreviation.

19) Line 426 – TIA was used for automatic data acquisition. This is surprising, how was it done.

To get raw data from Falcon III camera, we have to use TIA instead of SerialEM.

When acquiring data, the coordinates of sample position were read and stage was moved by scripts of SerialEM. A script of a mouse emulator was used to click the “Acquire” button on microscope control panel. The movies were recorded and stored by TIA.

20) Line 430 – it is mentioned that 96,000x @ 300 kV corresponds to 1.06 Å and 1 Å at 120 kV. But, if I remember right, the pixel size of Krios at 96,000 x is around 0.85 Å. Please check these. (Also, all the pixel values at 120 kV are lower than at 300 kV and only this is larger). In line 454, it is mentioned that 0.85 Å was used but not sure if this was apoferritin data mentioned in line 430.

We thank the reviewer for pointing out the error in the nominal magnification. The nominal magnification of apo-ferritin acquire by Falcon III is indeed 75,000x. We also corrected the nominal magnification of Cocksackievirus A10 dataset with Falcon III to 59,000x. The table below shows the nominal magnifications and corresponding pixel sizes of the Titan Krios microscope we measured.

Magnification	Pixel size of K3	Pixel size of Falcon III
22,500x	1.07Å@300kV, 1.03Å@160kV, 1.03Å@120kV	
47,000x	0.52Å@300kV	1.70Å@200kV
59,000x	0.39Å@120kV	1.36Å@300kV, 1.31Å@120kV

75,000x		1.06Å@300kV, 1.00Å@120kV
96,000x		0.86Å@120kV

21) Would it be possible to mention the gun lens (and or extraction voltage) used for different voltages. I guess the gun lens was higher at lower voltage.

The gun lens is 5 for 120/160 kV data acquisition and 6 for 200/300 kV. The extraction voltage was 4,500 V, which was unchanged during experiments.

22) Figure legend 1 – line 606, correct the sentence to ‘image was taken’

We thank the reviewer for the suggestion and have changed the words.

Supplementary material:

Supplementary note 2: line 313 – change to ‘data was acquired at a magnification ..’

We thank the reviewer for the suggestion and have changed the words.

In supplementary note 6 – in several places (line 400,401,413) formula is mentioned but it is not clear which formula it is being referred to. It just says supplementary 3 or 5 or 6. But there is none in supplementary 5. I think it should Eq.2, Eq.5 and so on.

We thank the reviewer for the suggestion and have changed the markers to Eq.x or

Supplementary Eq.x.

Line 411, please change implied to implemented

We thank the reviewer for finding the typo and we have corrected it.

Reviewer #2 (Remarks to the Author):

The manuscript by Zhu et al. propose a new counting algorithm for lower electron energies (200 keV and 120 keV) that is claimed to be able to surpass the performance of the most widely used 300 keV instrumentation, particularly for small proteins (<100 kD in molecular mass). Despite the technical breakthroughs in hardware and software for biological cryo-EM studies, structural determination of small biomolecules remains very challenging as the authors rightly pointed out. However, the manuscript is quite challenging for non-experts with too many technical jargons that are not well explained. Most importantly, the authors failed to present a real biological case of which the molecular mass is sufficiently small to demonstrate the advantage of the proposed algorithm. Instead, the test cases are big with high symmetries. It is not trivial to evaluate how the proposed algorithm will actually overcome the size limitations in a real world. This reviewer therefore cannot recommend the publication of this manuscript in Communications Biology.

Specific comments are as follows:

We thank the reviewer for the comments. We have stated in the previous manuscript that “We compared the latter with the actual FSC obtained at 120 kV (Fig. 4c and d). Clearly, the Cc was the main limitation, and the resolution of the 3D reconstruction at 120 kV is no worse than that at 300 kV if the Cc envelope depression effect was eliminated.” We did not claim that the low-acceleration-voltage TEM surpass the performance of the most widely used 300 keV instrumentation. Indeed, we have stated in the manuscript that “chromatic aberration is a major limiting factor in low-voltage cryo-EM” . In addition, we have revised the manuscript according to the suggestions from all reviewers. The revised manuscript can be understood better for readers. We collected new dataset and analyzed synthesized datasets on small proteins. Our results showed better performance of our new algorithm on smaller proteins. The detailed results were shown below.

1. The advantage of using a lower electron energy has been discussed by Chris Russo and colleagues in 2019 (ref. 15) in which a wide range of molecular masses was tested to demonstrate the advantages of collecting data at 100 keV. While this manuscript provides a great deal of technical details by benchmarking against different instrumentation setups and data processing algorithms, it appears to miss the most important point that such a development should indeed help improve the data quality of small protein systems. The authors should use some model systems that have been evaluated before, such as hemoglobin (64 kD), to demonstrate the

improved performance.

As we mentioned in the manuscript, our new counting algorithm improves the SNR by 20% at half the Nyquist frequency and 80% at Nyquist frequency when implemented in a Falcon III camera to 120keV electrons. For cryo-EM data of a 30nm-diameter Coxsackievirus, the large sample produced enough contrast and images produced by traditional counting algorithm contained enough signal for an accurate alignment in the refinement of the reconstruction, thus the improved images reproduced by our new counting algorithm did not help improve the alignment in this case. The increased SNR of cryo-EM images only led to a resolution improvement of 0.17 Å. However, for apo-ferritin, a 450kD protein, such an improvement in SNR of cryo-EM images by our new counting algorithm led to a better alignment, which improved the resolution of 0.52 Å. These results indicated that smaller proteins are benefited more from the new counting algorithm. However, the envelope functions of a TEM with different acceleration energies as we calculated in the manuscript (Supplementary Fig. 9a) shows that the SNR at frequency of $1/5 \text{ \AA}^{-1}$ drops ~35% in a 120 kV TEM compared with a 300 kV TEM. As we mentioned in the manuscript, such a decreasing of SNR due to the envelope function is the main shortcoming of the low-acceleration-voltage TEM. Streptavidin, a 52kD protein is also a model system to test the ability of solving small proteins in cryo-EM. The size of the protein is so small that a close-to-perfect cryo-EM specimen preparation is required to produce cryo-EM images with best SNR for a successful reconstruction. The decreasing of SNR at high frequency in a

low-acceleration-voltage TEM may result in misalignment. Indeed, we collected cryo-EM data of Streptavidin at both 120 kV with Falcon III and 300 kV with K3 in a Titan Krios. We showed a 120kV cryo-EM image with defocus of $-1.253 \mu\text{m}$ and a 300kV cryo-EM image with defocus of $-2.035 \mu\text{m}$ here. The defocus was selected to make sure a similar CTF function at low frequencies for both images. The images were low-pass filtered to a frequency of $1/20 \text{ \AA}^{-1}$ and exhibit similar contrast. However, the results of the fitting of CTF by CTFFIND4 showed that the detectable Thon ring extended to only $1/5 \text{ \AA}^{-1}$ on the 120-kV image, while it extended to $1/3.5 \text{ \AA}^{-1}$ on the 300-kV image. These results agreed with the envelope functions.

120kV, defocusU/V=-1.350/1.156 μ m,
Thon ring fit good up to 5.05 \AA

Both images were displayed using relion_display with scale=0.2, sigma contrast=3, low-pass=20 \AA

300kV, defocusU/V=-2.051/2.018 μ m,
Thon ring fit good up to 3.50 \AA

The further 2D classification results by RELION were shown below. At 300 kV, two major side-views were reported to achieved 3.7 \AA and 4.2 \AA , respectively. These side-views, however, were very blur and reported resolution dropped to 7.2 \AA and 6.4 \AA at 120 kV. Top-views which were marked in red showed more glitch at 300 kV than 120 kV, and the reported resolution was the same for a typical class of top-view.

7.19Å

6.60Å

5.69Å

Three typical views at 120 kV

3.71Å

4.19Å

5.69Å

Three typical views at 300 kV

However, 3D reconstruction failed for 120-kV dataset with both RELION and cryoSPARC. The final resolution of apo-streptavidin at 300kV was 3.2 Å reported by RELION, with ~400 micrographs and ~110,000 particles. At 120 kV, although 3D auto-refinement was forced, the final resolution was 7.1 Å and the reconstructed map did not present any detail of structure.

To further examine the performance of our new counting algorithm on small proteins, we synthesized non-symmetric proteins with different sizes by subtracting partial densities from apo-ferritin dataset as shown in figure below. During the process, densities outside of the masks were subtracted from the 2D images and densities inside the mask were subjected to a non-symmetrical local refinement in the 120-kV apo-ferritin dataset.

Four datasets were synthesized corresponding to molecular weights of 340 (3/4), 225 (1/2), 113 (1/4) and 75kD (1/6 of total mass), respectively. Since the density subtraction was not perfect, 3D masks as shown in the figure excluding the subtracted densities in the 3D references were used during the refinement of the structures. In addition, only local refinement was performed on all the subtracted datasets. The local refinement was a “continual running” after the 3rd iteration of auto-refinement. For synthesized proteins with different sizes, we calculated the $SNR_{(0.25)}$ of the 3D reconstructions at frequency of $1/4 \text{ \AA}^{-1}$ based on FSC curve. The improvement of new counting algorithm “Hybrid” was calculated by divided $SNR_{(0.25)}$ of the reconstruction from traditional counting “Mass-Centre” images by $SNR_{(0.25)}$ of the reconstruction from new counting images. The $SNR_{(0.25)}$ improved 49.9, 46.3, 86.0 and 123.9% in synthesized non-symmetric proteins with size of 340, 225, 113 and 75 kD by linear fitting results. Results show that with smaller mass of

protein, the SNR of our new counting method improved more significantly versus conventional counting method.

We also tested a synthesized non-symmetric protein with the size of 56kD (1/8 of total mass). However, the protein is probably too small for alignment, resulted in a dramatic drop of resolution as shown on the slices of the reconstruction below.

2. There are several technical terms that require better definitions and explanations. For example, the terms PEEK and GC (line 200) were not defined throughout the manuscript.

We thank the reviewer for the suggestion and have expanded the abbreviation.

3. Even for the selected model systems that are large and highly symmetric, the

authors did not provide the actual cryo-EM maps derived from different algorithms together with the atomic models, making it difficult to judge the performances simply based on the FSC curves.

We thank the reviewer for the suggestion and we have added Supplementary Figure 12 to show the density of residue and the corresponding atomic model for apo-ferritin dataset at 120 kV with our new counting algorithm and conventional counting. As for the rest datasets with our counting algorithm, we have deposited the maps to the EM databank.

4. As a minor comment, many plots have very small labels, which limited the readability.

We thank the reviewer for the suggestion and we have changed the size of each label of our plots to let viewers read better.

Reviewer #3 (Remarks to the Author):

The authors present a very welcome new electron counting algorithm for CryoEM, especially well-suited to 100 and 200 kV. The algorithm is sound and its evaluation very thorough. The article is generally well written and there are only a few aspects that could be improved:

We thank the reviewer for the comments.

- One wonders if the parameter pow can be considered a continuous variable, instead of discrete (0, 1, 2, ...) and better results would be obtained.

We thank the viewer for the suggestion. In factor we had already treat pow as a continuous variable. We had tested values such as ± 0.25 , ± 0.5 , ± 0.75 , ± 1.0 , ± 1.25 , ± 1.5 Those results were consistent with results we showed. We only showed some of the value to make graphs clearer.

- Suppl. Table 1. It would be good if the input data is shown and the R^2 of the regression reported.

We thank the viewer for the suggestion. We have added the R-square in Supplementary Table 1. The fitted vectors of screen and K3 camera is shown below,

where the sigma value represents the standard error (in pixel) between two groups of vectors.

- It is unclear why they needed to modify Relion3. Now that Relion4 is out, how portable are the changes?

During our experiments, RELION4 was not published so we had to modify RELION3.

RELION polish contains three major steps. First one is to calculate per-particle frame-shift (stored in "tracks.star" file) in movie. Second one is to calculate B-factor like SNR weighting filters (stored in three "FCC" files). Third is to read those files and generate particle images. Since we change the input images, original version of RELION will re-calculate shift and weighting files, thus affects the SNR of particle images. After modification of RELION3, we ensure that RELION polish only reads given files in electron counting tests or ignores per-particle translational alignment and SNR weighting filters in critical dose measurements when generating particle images. In such a way, the SNR of reconstructions can be directly compared.

We only changed several functions in RELION3 and a modified version of RELION4 would be provided if needed.

- The EMDB maps have no public code.

We thank the viewer for the suggestion and we have deposited the EM density maps in the Electron Microscopy Data Bank under accession codes EMD-32603/32604/32605 (Hybrid, MC and WP counting of Coxsackievirus A10 at 120 kV), EMD-32606/32607/32608 (MC, PV and PCA counting of large-sized clusters of Coxsackievirus A10 at 120 kV), EMD- 32609/32611 (MC, Hybrid counting of Coxsackievirus A10 at 200 kV), EMD-32600/32601/32602 (120/160/300 kV Coxsackievirus A10 from critical dose measurement), EMD-32610 (PCA counting of apo-ferritin at 300 kV), EMD- 32612/32613 (Hybrid, MC counting of apo-ferritin at

120 kV).

REVIEWERS' COMMENTS:

Reviewer #1 (Remarks to the Author):

The revised manuscript by Zhu et al describing a new counting algorithm for the use with lower voltage(s) with existing direct electron detectors has addressed much of the queries raised by reviewers. This includes the comment on the use of the new algorithm for smaller proteins (50-60 kDa). The text and flow has improved but it still has some sentences with language issues, which I presume can be addressed during proof.

There is one clarification required from the authors – the table they had added in the revised manuscript describing the data sets collected and parameters (Lines 775-777). In one of this – line 776 (or 777), at 200 kV and 47,000, the pixel size is 1.7 Å but the resolutions of the map using both MC and hybrid counting is 2.72 Å and 2.66 Å. But the Nyquist limit is 3.4 Å.

The reference to this in the text is (line 299-302) – for 200-kv dataset, we used 2x super-resolution mode for both hybrid and MC (and the supplementary figure S6 -f). I am assuming the actual pixel size is 0.85 Å in super-resolution mode (?). Please clarify.

Line 466 – R12/13 – should be R1.2/1.3

Reviewer #2 (Remarks to the Author):

The authors have substantially revised the manuscript according the reviewers' comments and suggestions. As far as the technicality is concerned, the descriptions on the data process algorithm are now much improved. With regard to the practical applications on smaller systems, however, the authors presented 2D classification data of streptavidin at both 120 and 300 keV, but fell short in elaborating on the performance of the new algorithm for really small protein systems (<100 kDa in the overall size). Instead, the authors presented a benchmark using synthetic datasets that extract partial EM images from apoferritin, which was rather misleading in my opinion. The authors are suggested to downplay the statement in the first sentence of the abstract. While it is technically correct in stating that one of the challenges in studying small protein systems by cryo-EM is the low SNR, the authors concluded in the rebuttal letter but did not disclose the same findings in the revised manuscript that the EM images of streptavidin collected at 120 keV could not be reliably aligned to yield satisfying 3D maps. Likewise, the closing sentence of the abstract suggested that this study is expected to provide solutions to image small proteins using a 120 keV cryo-TEM instrument, but the actual limitation in the workable protein size has not been appropriately demonstrated or discussed in this study. Having said that, the technical advancements made in this study are important findings for specialized developers in the cryo-EM community.

Reviewer #3 (Remarks to the Author):

The authors have significantly improved the article and it can now be published.

REVIEWERS' COMMENTS:

Reviewer #1 (Remarks to the Author):

The revised manuscript by Zhu et al describing a new counting algorithm for the use with lower voltage(s) with existing direct electron detectors has addressed much of the queries raised by reviewers. This includes the comment on the use of the new algorithm for smaller proteins (50-60 kDa). The text and flow has improved but it still has some sentences with language issues, which I presume can be addressed during proof.

We thank the reviewer for the comments of the manuscript.

There is one clarification required from the authors – the table they had added in the revised manuscript describing the data sets collected and parameters (Lines 775-777). In one of this – line 776 (or 777), at 200 kV and 47,000, the pixel size is 1.7 but the resolutions of the map using both MC and hybrid counting is 2.72 Å and 2.66 Å. But the Nyquist limit is 3.4 Å.

At 47,000x, the physical pixel size of FalconIII was 1.7 Å. We used Hybrid filter in 2x2 super-resolution mode described on Supplementary Note 6 to overcome Nyquist limit. As a result, the final pixel size of our 200 kV reconstruction is 0.85 Å.

The reference to this in the text is (line 299-302) – for 200-kv dataset, we used 2x super-resolution mode for both hybrid and MC (and the supplementary figure S6 -f). I am assuming the actual pixel size is 0.85 Å in super-resolution mode (?). Please clarify.

We have rewritten the Method part and modified the sentences into “320 by 320 pixels with pixel size of 1.7 Å in non-super-resolution mode and 640 by 640 with pixel size of 0.85 Å in 2x2 super-resolution mode” (line 510-511) and “2.66 Å with super-resolution at 200 kV and 2.88 Å at 120 kV for Cocksackievirus A10 on the basis of the gold-standard FSC = 0.143 criterion.” (Line 519-521), to clarify the issue.

Line 466 – R12/13 – should be R1.2/1.3

We thank the reviewer for finding the problem. That name of GIG grid was “R12/13” shown in the production catalog, meaning R1.2/1.3. For conventional understanding, we have changed the name to “R1.2/1.3”.

Reviewer #2 (Remarks to the Author):

The authors have substantially revised the manuscript according the reviewers' comments and suggestions. As far as the technicality is concerned, the descriptions on the data process algorithm are now much improved. With regard to the practical applications on smaller systems, however, the authors presented 2D classification data of streptavidin at both 120 and 300 keV, but fell short in elaborating on the performance of the new algorithm for really small protein systems (<100 kDa in the overall size).

We have added the sentence “Further 3D-reconstruction of the 120-kV dataset also led to misalignment of images and cannot provide any detailed structure.” (Line 363-364)

Instead, the authors presented a benchmark using synthetic datasets that extract partial

EM images from apoferritin, which was rather misleading in my opinion. The authors are suggested to downplay the statement in the first sentence of the abstract. While it is technically correct in stating that one of the challenges in studying small protein systems by cryo-EM is the low SNR, the authors concluded in the rebuttal letter but did not disclose the same findings in the revised manuscript that the EM images of streptavidin collected at 120 keV could not be reliably aligned to yield satisfying 3D maps. Likewise, the closing sentence of the abstract suggested that this study is expected to provide solutions to image small proteins using a 120 keV cryo-TEM instrument, but the actual limitation in the workable protein size has not been appropriately demonstrated or discussed in this study.

Having said that, the technical advancements made in this study are important findings for specialized developers in the cryo-EM community.

As suggested by Peet et al. (2019), lowering the acceleration voltage enhances signals of smaller proteins substantially (<100kD). They estimated the radiation damage by measuring intensity of diffraction spots on 2D-crystals. Here, we re-estimated the radiation damage by measuring critical doses on a protein by single particle analysis. As discussed in the manuscript, our result is different from that estimated from 2D crystals, but close to the theoretic result from Estar database. Thus, our results confirmed their prediction based on the data from Estar database that lowering the acceleration voltage enhances signals of smaller proteins substantially. These results had been detailed described in the early version of the manuscript. However, to highlight the counting algorithm dedicated to low energy electrons, we have reduced this part before the submission of the manuscript. Thus, we agreed the reviewer and have modified the abstract to "Relative to the 300-kV accelerating field, electrons accelerated under lower voltages are potentially scattered more strongly. Lowering the accelerate voltage has been suggested to enhance the signal-to-noise ratio (SNR) of cryo-electron microscopy (cryo-EM) images of small-molecular-weight proteins (<100 kD). However, the detection efficient of current Direct Detection Devices (DDD) and temporal coherence of cryo-EM decrease at lower voltage, leading to loss of SNR."

As we have shown in Figure 4c and 4d, in addition to a perfect detector, the chromatic aberration is the major disadvantage of a low acceleration voltage microscope and without the chromatic aberration, the resolution from the 120kV microscope is comparable to that of a 300kV microscope. Based on the above analysis, the final sentence of Abstract that "Our results indicate that with further improved temporal coherence and a dedicated designed camera, a 120-kV cryo-electron microscope has potential to match the 300-kV microscope at imaging small proteins." describes our results and we decide to keep that.

Reviewer #3 (Remarks to the Author):

The authors have significantly improved the article and it can now be published.

We thank the reviewer for reviewing our manuscript.